# Advancing Fine-Grained Classification by Structure and Subject Preserving Augmentation

**Eyal Michaeli**
Department of Computer Science
Reichman University
eyal.michaeli@post.runi.ac.il

**Ohad Fried**
Department of Computer Science
Reichman University
ofried@runi.ac.il

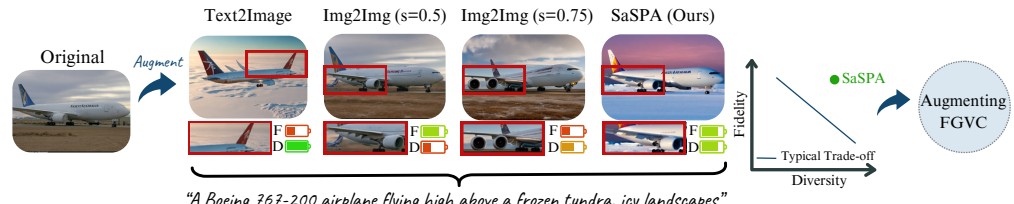

Figure 1: Various generative augmentation methods applied on Aircraft [30]. Text-to-image often compromises class fidelity, visible by the unrealistic aircraft design (i.e., tail at both ends). Img2Img trades off fidelity and diversity: lower strength (e.g., 0.5) introduces minimal semantic changes, resulting in higher fidelity but limited diversity, whereas higher strength (e.g., 0.75) introduces diversity but also inaccuracies such as the incorrectly added engine. In contrast, SaSPA achieves high fidelity and diversity, critical for Fine-Grained Visual Classification tasks. *D - Diversity. F - Fidelity*

## Abstract

Fine-grained visual classification (FGVC) involves classifying closely related sub-classes. This task is difficult due to the subtle differences between classes and the high intra-class variance. Moreover, FGVC datasets are typically small and challenging to gather, thus highlighting a significant need for effective data augmentation. Recent advancements in text-to-image diffusion models offer new possibilities for augmenting classification datasets. While these models have been used to generate training data for classification tasks, their effectiveness in full-dataset training of FGVC models remains under-explored. Recent techniques that rely on Text2Image generation or Img2Img methods, often struggle to generate images that accurately represent the class while modifying them to a degree that significantly increases the dataset's diversity. To address these challenges, we present SaSPA: Structure and Subject Preserving Augmentation. Contrary to recent methods, our method does not use real images as guidance, thereby increasing generation flexibility and promoting greater diversity. To ensure accurate class representation, we employ conditioning mechanisms, specifically by conditioning on image edges and subject representation. We conduct extensive experiments and benchmark SaSPA against both traditional and recent generative data augmentation methods. SaSPA consistently outperforms all established baselines across multiple settings, including full dataset training, contextual bias, and few-shot classification. Additionally, our results reveal interesting patterns in using synthetic data for FGVC models; for instance, we find a relationship between the amount of real data used and the optimal *proportion* of synthetic data. We release our source code.

38th Conference on Neural Information Processing Systems (NeurIPS 2024).

# 1 Introduction

Deep learning's remarkable success across various applications relies heavily on large-scale annotated datasets, such as ImageNet [12], which provide the foundational data necessary for training effective models. However, in fine-grained visual classification (FGVC), the datasets are typically smaller and less diverse, presenting unique challenges in training robust models. Data augmentation emerges as a natural solution to artificially enhance dataset size and variability. However, traditional data augmentation methods are limited in the amount of diversity they introduce [15].

Text-to-image diffusion models have opened new avenues for generative image augmentation. Within the realm of classification, diffusion models have shown promise on standard image recognition datasets such as ImageNet [2, 50, 3]. However, their application in FGVC remains under-explored.

Generating synthetic data for FGVC presents unique challenges, as preserving class fidelity is (1) more crucial than with common object datasets due to the similarity between classes and the reliance of the models on subtle details to differentiate between classes and (2) challenging to achieve because the training data for text-to-image models often lacks a substantial representation of these distinct objects [26]. For instance, there might be enough data to accurately represent "An airplane", but not "A Boeing 767-200 airplane".

Recent generative methods evaluated for FGVC augmentation typically use real images as guidance in an Img2Img manner [15, 54, 21, 60]. While this helps maintain visual similarity to the target domain, it limits the degree of diversity that can be introduced, resulting in a trade-off between class fidelity and diversity [16] (see Figure 1). We aim to free the generative process from this constraint of adhering to specific source images. To this end, we propose SaSPA: Structure and Subject Preserving Augmentation, a method that conditions the generation on more abstract representations rather than direct image inputs. Specifically, we leverage structural conditioning in the diffusion model via edge maps extracted from source images. This allows the generated samples to respect the broad shape and composition of objects in the target domain. Crucially, the lack of specific image conditioning enables greater flexibility in rendering surface details. To further ensure the preservation of fine-grained class characteristics, we integrate subject representation conditioning. By combining edge-based structural conditioning with category-level conditioning, SaSPA can generate highly diverse, class-consistent synthetic images without being overly influenced by any specific real data sample.

Furthermore, to enrich the diversity and applicability of our generated images, we generate prompts with an LLM according to the dataset meta-class (a class encompassing all sub-classes). These prompts are designed to guide the diffusion model in producing variations that are not only diverse but also class-consistent and relevant to the target domain. Additionally, to maintain the quality and relevance of the generated images, we implement a robust filtering strategy that eliminates any samples that fail to meet predefined quality thresholds by utilizing a dataset-trained model and CLIP.

**We summarize our contributions as follows**: (1) We propose SaSPA, a generative augmentation pipeline for fine-grained visual classification that generates diverse, class-consistent synthetic images without relying on specific real images for conditioning. (2) We conduct extensive experiments and benchmark SaSPA against both traditional and recent generative data augmentation methods. SaSPA consistently outperforms all established baselines across multiple settings, including the challenging and less-explored full dataset training, as well as in scenarios of contextual bias and few-shot classification. (3) Our analysis provides insights on effectively leveraging synthetic data to improve the performance of fine-grained classification models. For instance, we find that as the amount of real data decreases, we should increase the *proportion* of synthetic data used.

# 2 Related Work

**Data Augmentation with Generative Models.** Synthesizing training samples using generative models is an active and challenging area of research. Initial efforts in this field [70, 4, 49, 33] leveraged Generative Adversarial Networks (GANs) to create labeled training samples. Recently, the emergence of powerful text-to-image diffusion models such as Stable Diffusion [46] has created exciting opportunities for advancing generative image augmentation. These models have been employed across a range of applications, including semantic segmentation [19, 61, 62, 36], object detection [8, 7, 59], and classification [34, 2, 50, 3], demonstrating their versatility and effectiveness For image classification tasks, diffusion models have demonstrated promising results on standard

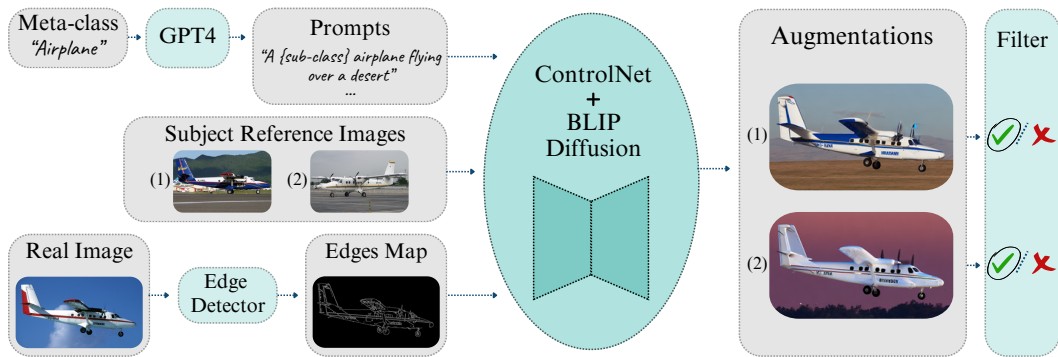

Figure 2: **SaSPA Pipeline:** For a given FGVC dataset, we generate prompts via GPT-4 based on the meta-class. Each real image undergoes edge detection to provide structural outlines. These edges are used $M$ times, each time with a different prompt and a different subject reference image from the same sub-class, as inputs to a ControlNet with BLIP-Diffusion as the base model. The generated images are then filtered using a dataset-trained model and CLIP to ensure relevance and quality.

image recognition datasets such as ImageNet [2, 50, 3]. However, their application in FGVC has typically been limited to particular settings such as few-shot learning [21, 54, 52, 26] where data scarcity significantly enhances the impact of data augmentation, contextual bias, and domain generalization [15], settings that are more straightforward to enhance as targeted augmentations can directly address and balance the skewed distributions. Our goal is to tackle the more challenging task of training on full FGVC datasets. Moreover, recent generative augmentation methods often use Img2Img techniques like SDEdit to maintain class fidelity, though this comes at the cost of reduced diversity. Some methods involve fine-tuning the network or its components, which can be expensive and may still struggle to balance class fidelity with the added diversity necessary for effective FGVC augmentation. Our goal is to avoid the decrease in diversity associated with using real images as guidance and to avoid the complexity and expense of fine-tuning the generation model.

**Text-to-Image Diffusion Models.** Diffusion models [23] have achieved unprecedented success in generating photo-realistic images [13]. Models like Stable Diffusion [46], DALL-E 2 [43], and others [37, 48] exemplify this capability. These models have also driven advancements in other generative areas. For instance, SDEdit [32] integrates real images partway through the reverse diffusion process for image editing. Techniques like ControlNet [68] and T2I-Adapter [35] condition image generation on inputs beyond text such as edges and world normals, while methods such as Textual Inversion [18] and DreamBooth [47] can generate specific subjects from just a few example images. More recently, BLIP-diffusion [27], which is based on Stable Diffusion and BLIP-2 [28], has demonstrated impressive zero-shot subject-driven generation using only one example image. Our method benefits directly from these advancements, employing ControlNet and BLIP-diffusion.

**Traditional Data Augmentation.** Traditional data augmentation methods typically include operations such as random cropping, flipping, and color-space changes to generate new variations [10]. Recent strategies, like mixup-based methods, aim to enhance diversity by blending patches from two input images [66] or using convex combinations [67]. Weakly Supervised Data Augmentation Network (WS-DAN), used in recent FGVC works such as CAL [44], aims to improve FGVC by generating attention maps to highlight discriminative object parts and guiding augmentation with attention cropping and dropping. However, these methods introduce limited diversity [15], as they do not alter the semantic features present in the image.

## 3 Method

Our goal is to augment a labeled training dataset for FGVC to increase its diversity while faithfully representing the sub-classes. The key insight of our method is to minimize reliance on any particular source image during generation and instead condition the generation on more abstract representations, thereby increasing diversity while accurately representing the designated class (see Figure 1). To achieve this, we employ abstract conditions such as edges, which capture the object's structure,

and subject representation, which aims to preserve fine-grained class characteristics. The process, illustrated in Figure 2, unfolds in five steps, outlined below:

## 3.1 Construction of Prompts

We aim to use a generative text-to-image model, which requires input prompts to guide the image synthesis process. To ensure that the prompts generated align broadly with the primary category of each dataset, our method begins by identifying the *meta-class* for each dataset, such as "Airplane" for the Aircraft [30] dataset and "Car" for the Stanford Cars [24] dataset.

**Prompt Generation via GPT-4 [1].** We input the meta-class to GPT-4 with the instruction to produce 100 unique, relevant, and diverse prompts, each inherently containing the term of the meta-class. This strategy ensures that the generated images stay true to the fundamental aspects of the meta-class while hopefully containing relevant and diverse scenarios. To increase the specificity and relevance of these prompts, we integrate the relevant sub-class into each prompt whenever it is used. Unlike a recent work [15], which also uses GPT-4 to create prompts, we do not require image captions of the dataset. The exact instructions for GPT-4 and more example prompts are in Appendix E.1.

## 3.2 Visual Prior Extraction

To ensure that the generated images maintain the overall structure and shape of objects belonging to their respective classes, we condition the synthesis process on edge representations extracted from real images in the dataset. Concretely, for a dataset of $N$ images, we extract one-channel edge representations using the Canny edge detector [6]. This yields a set of $N$ edge-based visual priors that capture the structural characteristics of each sub-class. By conditioning the generative model on these edge maps, we can preserve object shape and layout during synthesis while allowing flexibility in rendering other surface-level details. In Table 17, we additionally explore the use of HED [63] as an alternative technique for edge extraction and structural conditioning.

## 3.3 Image Generation

Diversity in synthetic data is crucial for effective training [55, 45, 31]. Unlike most recent approaches [21, 15, 54], our method focuses on edges and subject representation as a prior rather than the source image. We show in Figure 1 that this approach maintains class fidelity, and as a result of not using the source image, affords the model greater flexibility to introduce novel semantic features such as weather, lighting, or even new elements both within and outside the object's confines. This strategy might be particularly beneficial in FGVC tasks, where the subtle differences between classes are crucial, and hence, maintaining class fidelity while introducing diversity is of paramount importance.

**Conditioning on edge maps** of a real image ensures that generated images align closely with real structural features. Interestingly, we notice that structural conditioning cues the generation model to accurately represent the target sub-class, enhancing not only the correct representation of structural features but also the correct representation of non-structural attributes like color and texture. This structure-guided synthesis approach effectively enhances the model's ability to maintain class fidelity across varied image generations.

**Conditioning on subject representation** further enhances the generation model's ability to produce images with accurate sub-class representation. This ensures the correct representation of the sub-class on levels beyond structure, such as texture, color, and other visual features.

Using these two mechanisms together ensures correct sub-class representation across datasets, whether the primary distinctions between sub-classes lie in structural features, texture, color, or any combination of them.

Due to its impressive results and widespread use, we employ ControlNet [68] conditioned on Canny edges [6] for edge map conditioning. For subject representation conditioning and as a base model for ControlNet, we utilize BLIP-diffusion [27], a model built upon Stable Diffusion [46] and BLIP-2 [28] that emphasizes subject representation and supports zero-shot subject-driven generation using one reference image. We chose BLIP-diffusion for its zero-shot capabilities and open-source availability.

Using BLIP-diffusion with ControlNet requires a prompt, an edge map, and a reference image of the target subject. To maintain sub-class accuracy while introducing *diversity* at the sub-class level, the

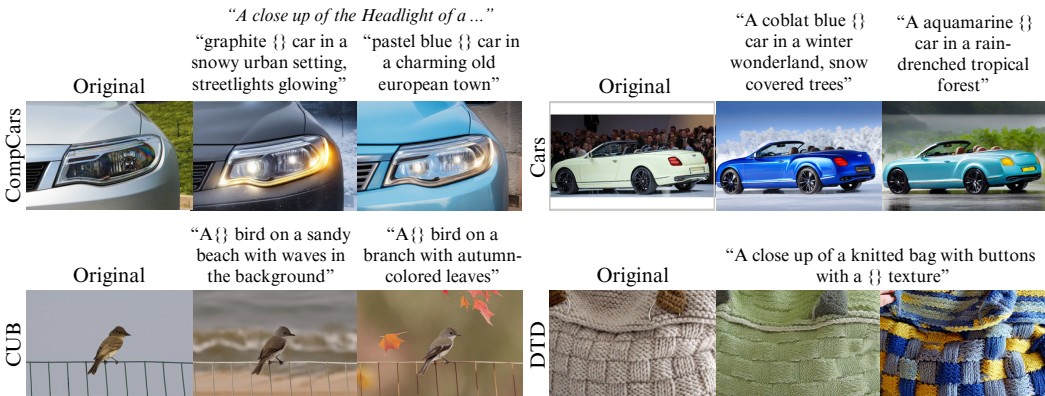

Figure 3: Example augmentations using our method (SaSPA). The {} placeholder represents the specific sub-class.

reference image is selected from the same sub-class but *differs* from the real image used to extract the edge map (we experiment with BLIP-diffusion inputs in Table 4). Specifically, we generate $M = 2$ augmentations for each real image in the training set: we extract an edge map for each real image, and for each edge map, we randomly select $M$ prompts and $M$ subject reference images from the same sub-class. These inputs are then fed into the generation model of ControlNet with BLIP-diffusion as the base model to produce $M$ augmentations of the real image. Example augmentations are visualized at Figure 3. DTD [9] has only one prompt per real image, because we utilize image captions as prompts for it, as explained in Appendix D.

## 3.4 Filtering

We aim to remove low-quality augmentations, which appear in two forms: (1) *meta-class* misrepresentation and (2) *sub-class* misrepresentation.

**Semantic Filtering**. To alleviate *meta-class* misrepresentation, we utilize semantic filtering as described in ALIA [15]. Using CLIP [41], this process evaluates the relevance of generated images to the specific task at hand. For example, in a car dataset, each generated image is assessed against a variety of prompts such as "a photo of a car", "a photo of an object", "a photo of a scene", "a photo", and "a black photo". Images that CLIP does not recognize as "a photo of a car" are excluded to ensure that the augmented dataset closely aligns with the target domain.

**Predictive Confidence Filtering**. To ensure each augmentation faithfully represents its designated *sub-class*, we implement a predictive confidence filtering strategy inspired by recent work [21] strategy *CLIP Filtering*. This method employs CLIP [41] to filter out images that do not strongly correlate with the textual labels of their class among all classes in the dataset. However, the limitation of using CLIP in this context is its insufficient granularity in understanding fine-grained concepts. For our method, we discard any augmented images for which the true label does not rank within the top-k predictions of a baseline model trained on the original dataset. This approach helps to exclude images that likely misrepresent the source *sub-class*, thus maintaining a high-quality dataset for model training. In our implementation, we use $k = 10$. Further details about this method, the baseline model used, and other filtering techniques are discussed in Appendix E.2.

## 3.5 Training Downstream Model

We train the downstream classification model using the filtered, generated samples. Let $\alpha$ denote the augmentation ratio, representing the probability that a real training sample will be replaced with a generated synthetic sample during each epoch. This replacement process is repeated for every sample in each epoch, allowing each real sample to be either retained or replaced by an augmented version. We employ this replacement strategy instead of simply adding the augmented data to the original dataset, as doing so would unnecessarily increase the number of iterations per epoch. By that, we ensure fair comparisons across training sessions.

Table 1: **Results on full FGVC Datasets.** This table presents the test accuracy of various augmentation strategies across five FGVC datasets. The highest values for each dataset are shown in **bold**, while the highest validation accuracies achieved by traditional augmentation methods are underlined.

| Type | Augmentation Method | Aircraft | CompCars | Cars | CUB | DTD |
|------|---------------------|----------|----------|------|-----|-----|
| *Traditional* | No Aug | 81.4 | 67.0 | 91.8 | 81.5 | 68.5 |
| | CAL-Aug | 84.9 | 70.5 | 92.4 | 82.5 | 69.7 |
| | RandAug | 83.7 | 72.5 | 92.6 | 81.5 | 69.3 |
| | CutMix | 81.8 | 66.9 | 91.7 | 81.8 | 69.2 |
| | CAL-Aug + CutMix | 84.5 | 70.2 | 92.7 | 82.4 | 69.7 |
| | RandAug + CutMix | 84.0 | 72.6 | 92.7 | 81.2 | 69.2 |
| *Generative* | Real Guidance | 84.8 | 73.1 | 92.9 | 82.8 | 68.5 |
| | ALIA | 83.1 | 72.9 | 92.6 | 82.0 | 69.1 |
| *Ours* | SaSPA w/o BLIP-diffusion | **87.4** | 74.8 | 93.7 | 83.0 | 69.8 |
| | SaSPA | 86.6 | **76.2** | **93.8** | **83.2** | **71.9** |

# 4 Experiments

Our objective is to explore the extent to which synthetic data, particularly through our approach, contributes to various FGVC tasks. We aim to understand the significance of each component of our method and identify optimal strategies for leveraging synthetic data in FGVC.

## 4.1 Experimental Setup

For generation, we employ BLIP-diffusion for *SaSPA* and Stable Diffusion v1.5 [46] for all other diffusion-based augmentation methods.

For training, we follow the implementation strategy outlined in the CAL study [44], tailored for FGVC. We use ResNet50 [20] as the primary architecture within the CAL framework unless specified otherwise. Each dataset is fine-tuned using pre-trained ImageNet weights. More data generation and training details can be found in Appendix D.

**Comparison Methods.** We benchmark our method, *SaSPA*, against established traditional and generative data augmentation techniques. In the *traditional* category, our comparisons include: **CAL-Aug [44]:** utilizes random flipping, cropping, and color-space variations. **RandAugment [11]:** applies a series of random image transformations such as rotation, shearing, and color variations to training images. **CutMix [66]:** generates mixed samples by randomly cutting and pasting patches between training images to encourage the model to learn more localized and discriminative features. **Combined Methods:** Tests the synergistic effects of CAL-Aug with CutMix and RandAug with CutMix. In the *generative* category, we compare with: **Real-Guidance [21]:** applies Img2Img with a low translation strength ($s = 0.15$) to maintain high fidelity to the original images. **ALIA [15]:** Uses real image captions and GPT-generated domain descriptions based on these captions as prompts for Img2Img translations. Detailed descriptions of these baseline methods are in Appendix D.3.

## 4.2 Fine-grained Visual Classification

**Datasets.** We evaluate on five FGVC datasets, using the *full* datasets for training. We use Aircraft [30], Stanford Cars [24], CUB [58], DTD [9], and CompCars [64]. For datasets lacking a predefined validation split, we establish one. For CompCars, we utilize the exterior car parts split, focusing exclusively on classifying images of car components: head light, tail light, fog light, and front into the correct car type. Further details on the exact splits are provided in Appendix C.

**Results.** We present the test accuracy of various augmentation methods in Table 1. For each dataset, the most effective *traditional* augmentation method (marked by an underline) is identified using its validation set and consistently combined with all *generative* approaches to optimize performance for that dataset. This approach is grounded in findings that standalone *generative* methods generally perform better when integrated with *traditional* augmentations [60], a trend also evident in Table 7.

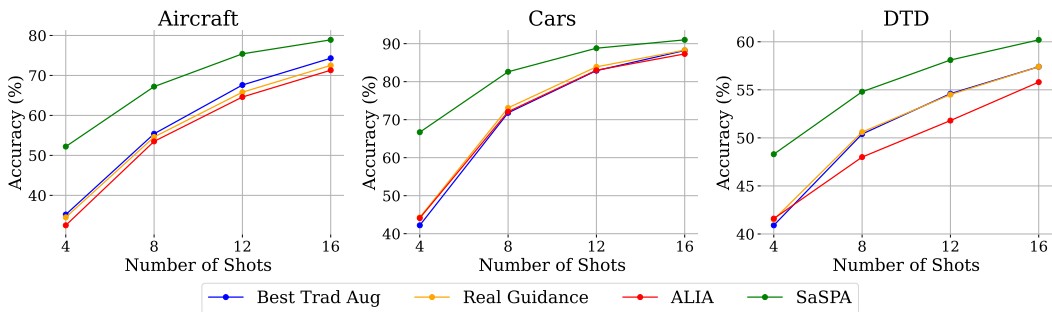

Figure 4: Figure 4: Few-shot test accuracy across three FGVC datasets: Aircraft, Cars, and DTD, using different augmentation methods. The number of few-shots tested includes 4, 8, 12, and 16. We can see that for all datasets and shots, SaSPA outperforms all other augmentation methods.

Our key findings: (1) *SaSPA* achieves consistent improvements across all datasets, with or without BLIP-diffusion integration, and it consistently outperforms traditional and generative augmentation methods by a significant margin. (2) The benefits of BLIP-diffusion vary depending on dataset characteristics; while it improves performance in datasets where texture and style play a crucial role in differentiation, such as DTD, CUB, and CompCars, it is not optimal for the Aircraft dataset, and has no significant impact on the Cars dataset, where structural features are more important for classification. We attribute this to the fact that using BLIP-diffusion confines the augmentations to be similar to other subjects within the same *sub-class*, which can limit diversity. (3) Both generative baselines fail to achieve consistent improvements and sometimes even reduce performance.

## 4.3 Few-shot Learning

**Experimental Setting**. This section investigates the efficacy of various augmentation strategies in few-shot fine-grained classification scenarios, focusing on how synthetic data affects performance with increasing numbers of training examples ("shots"). We conduct evaluations using three datasets: Aircraft [30], Cars [24], and DTD [9], assessing performance at 4, 8, 12, and 16 shots.

The training and data generation approaches remain consistent with those described in Appendix D, with two modifications: we use 100 epochs (down from 140), and we do not employ *predictive confidence filtering* for shots 4 and 8. The latter adjustment is due to the reduced reliability of the model's predictions, resulting from the limited training data. Additionally, we increase the augmentation ratio to $\alpha = 0.6$, as identified to be better for scenarios with limited data in Table 5.

**Results**. The results in Figure 4 show that *SaSPA* consistently outperforms all other augmentation methods across all datasets and various shot counts. As seen in other works [54, 21], the benefit of augmentation diminishes as the number of shots increases, a trend most noticeable in the Cars dataset. Contrary to prior work, the gains provided by *SaSPA* remain substantial even at higher shot counts; notably, in the Cars dataset at 16 shots, *SaSPA* achieves an accuracy of 91.0%, surpassing the second-best performance of 88.3% by RG [21]. Interestingly, *SaSPA* sometimes matches or exceeds the performance enhancement achieved by increasing the dataset size. For example, in the DTD dataset, utilizing *SaSPA* with 8 shots results in an accuracy of 54.8%, slightly surpassing the 54.6% obtained when adding 50% more real data (a total of 12 shots) when relying solely on real data and the best traditional augmentation.

## 4.4 Mitigating Contextual Bias (Airbus vs. Boeing)

**Experimental Setup.** To evaluate the effectiveness of our method in mitigating real-world contextual biases, we use the contextual bias split of the Aircraft [30] dataset constructed by Dunlap et al. [15]. The split uses two visually similar classes: Boeing-767 and Airbus-322. Each image in this split is categorized as "sky", "grass", or "road" depending on its background, with ambiguous examples filtered out. The bias in the dataset is introduced by training on 400 samples where Airbus aircraft are exclusively associated with road backgrounds and Boeing aircraft with grass backgrounds, although both types may appear against sky backgrounds. The exact split breakdowns are in Table 19.

Table 2: Classification performance on the contextually biased Aircraft dataset [30], detailing overall, in-domain (ID) and out-of-domain (OOD) accuracies for each augmentation method.

| Augmentation Method | Acc. | ID Acc. | OOD Acc. |
|---|---|---|---|
| Best Trad Aug (CAL-Aug) | 71.0 | **88.2** | 10.2 |
| Real Guidance [21] | 71.7 | 86.9 | 17.7 |
| ALIA [65] | 71.8 | 84.9 | 25.1 |
| SaSPA w/o BLIP-diffusion | **73.0** | 81.9 | **41.5** |

Table 3: Comparison to concurrent work *diff-mix* [60]. Test accuracy on 3 FGVC datasets. † indicates values taken from the diff-mix paper. *TI - Textual Inversion, DB - DreamBooth, ✗- No fine-tuning.*

| Aug. Method | FT Strategy | ResNet50@448 | | | ViT-B/16@384 | | |
|---|---|---|---|---|---|---|---|
| | | Aircraft | Car | CUB | Aircraft | Car | CUB |
| CutMix † | - | 89.44 | 94.73 | 87.23 | 83.50 | 94.83 | **90.52** |
| Diff-Mix † | TI+DB | 90.25 | 95.12 | 87.16 | 84.33 | 95.09 | 90.05 |
| Diff-Mix + CutMix† | TI+DB | 90.01 | 95.21 | **87.56** | 85.12 | 95.26 | 90.35 |
| SaSPA (Ours) | ✗ | 90.59 | 95.29 | 86.92 | 85.48 | 95.12 | 89.70 |
| SaSPA (Ours) + CutMix | ✗ | **90.79** | **95.34** | 87.14 | **85.72** | **95.37** | 89.92 |

We follow the same training and generation implementation settings as for the FGVC setting (Appendix D), and we compare against the same *generative* methods. We also compare against the optimal *traditional* augmentation for the Aircraft dataset (CAL-Aug), as defined in Section 4.2.

**Results.** The results in Table 2 show that *SaSPA* outperforms all other methods in overall and out-of-domain (OOD) accuracy, demonstrating its effectiveness in mitigating contextual bias. However, it falls short in in-domain (ID) accuracy. A distinct inverse relationship is observed between ID and OOD accuracy: methods that induce more significant changes from the original image—such as ALIA, which uses stronger translations than Real-Guidance (RG)—tend to achieve higher OOD accuracy but lower ID accuracy. This trend suggests that greater modifications can help reduce over-fitting to in-domain characteristics, enhancing the model's ability to generalize effectively to new, unseen conditions. As depicted in Figure 1, even a higher translation strength ($s = 0.5/0.75$) yields limited diversity compared to our method. Consequently, the alterations produced by RG and ALIA are insufficient to significantly mitigate the contextual bias present in the dataset, as effective background variation is crucial for addressing such biases.

## 4.5 Comparing *SaSPA* with Concurrent Work *diff-mix*

In this section, we compare our method with *diff-mix*, a generative augmentation approach proposed concurrently by Wang et al. [60]. *diff-mix* was also evaluated on full FGVC datasets and demonstrated impressive results. This method enriches datasets through image translations between classes, utilizing personalization techniques such as textual inversion [18] and DreamBooth [47] to fine-tune the generative model for each *sub-class*. This fine-tuning enhances the model's ability to capture and represent class-specific nuances. In contrast, our method does not involve fine-tuning, aiming to simplify the process and minimize computational costs.

**Experimental Setup.** In this analysis, we evaluate the performance of our *SaSPA* augmentation method using the *diff-mix* training setup, as detailed in their work. By using their open-source implementation, we further assess the robustness of our method with a different training setup. To ensure fairness, We use the same number of augmentations (M) as diff-mix did. More details regarding training setup are in Appendix B.3.

**Results**. Our results, detailed in Table 3, highlight where *SaSPA* performs well and identify areas for potential improvement. The findings can be summarized as follows: (1) While *diff-mix* employs computationally intensive fine-tuning techniques to enhance class representation, we prioritize simplicity and lower computational demands in our approach. Despite this, *SaSPA* consistently outperforms *diff-mix* on both the Aircraft and Cars datasets across all architectures, whether combined with CutMix or used alone, demonstrating its robustness across various augmentation contexts. This

Table 4: Ablation Study: Effects of different generation strategies on various FGVC Datasets. 'Subj.' means subject representation is used. 'Edges=Subj.' indicates that the real image used to extract the edges is the same as the subject reference image. 'Art.' indicates that half the prompts are *appended* with artistic styles. For each dataset, **bold** indicates the highest validation accuracy, and underline indicates the second highest. Ticks under each column mean the component is used.

| Method | Edge Guidance | Img2Img | Subj. | Inputs | Art. | Aircraft | Cars | CUB | DTD |
|---|---|---|---|---|---|---|---|---|---|
| Best trad aug | - | - | - | - | - | 84.3 | 92.7 | 81.4 | 67.9 |
| *Ours* | | | | | - | 83.3 | 92.9 | 82.1 | 67.8 |
| | | ✓ | | | - | 83.0 | 92.8 | 80.7 | 66.0 |
| | | | ✓ | | - | 81.5 | 91.6 | 81.1 | 68.1 |
| | ✓ | | | | - | 85.7 | 93.4 | 81.8 | 68.4 |
| | ✓ | | | | ✓ | **86.2** | 93.8 | 81.6 | 68.6 |
| | ✓ | ✓ | | | ✓ | 84.9 | 93.0 | 81.3 | 67.8 |
| | ✓ | | ✓ | Edges=Subj. | | 85.2 | 93.1 | 81.3 | 68.7 |
| | ✓ | | ✓ | Edges≠Subj. | ✓ | 85.5 | 93.7 | 82.6 | 69.2 |
| | ✓ | | ✓ | Edges≠Subj. | | 85.4 | **93.9** | **83.0** | **69.9** |

also shows that conditioning the generation on more abstract representations, as we do for correct class representation, can overcome the absence of extensive fine-tuning. (2) The CUB dataset posed unique challenges, with *diff-mix* outperforming *SaSPA* using ResNet50 and both *diff-mix* and *SaSPA* under-performing relative to CutMix using ViT-B/16. Notably, despite *SaSPA* outperforming all methods, including CutMix in Table 1, it does not perform as well here. We hypothesize that the use of higher resolution emphasizes finer details in each class, which may be overwhelming for *SaSPA* on some datasets but less so for *diff-mix*, likely due to the heavy fine-tuning process integrated into their method. These results indicate that some form of fine-tuning might be advantageous for complex datasets like CUB to achieve better performance. The full table, including comparisons to more augmentation methods, can be found in Appendix B.3.

## 4.6 Effect of Different Generation Strategies on Performance

In Table 4, we conduct an extensive ablation study to evaluate the effectiveness of our proposed generation strategies. Specifically, we examine the integration of edge guidance, Img2Img as an alternative for edge guidance with strength $= 0.5$ and in combination with edge guidance with strength $= 0.85$, and subject representation. We also investigate the effect of using the same image for both edges extraction and the subject reference image ("Edges=Ref."). Additionally, we test the impact of appending half of the prompts with artistic styles (column 'Art.') as described in Appendix B.8.

**The results** demonstrate the importance of combining structural and subject-level conditioning while enabling diverse generations through separate input sources, yielding the best performance across most datasets. Key observations include: (1) Edge Guidance alone improves performance significantly compared to Text-to-Image or SDEdit [32] (Img2Img), highlighting its important role in providing structural guidance. (2) Subject representation alone does not enhance performance, indicating additional structural conditioning is necessary. (3) Using different source images for edges and subject reference images adds beneficial diversity. (4) Surprisingly, text-to-image generation (first row) outperforms SDEdit, likely due to its increased diversity and despite the lower fidelity, which our filtering mechanism can handle as it filters out low-fidelity images. (5) Incorporating artistic prompts has inconsistent effects, usually boosting performance with Edge Guidance but often degrading it when combined with subject representation. This inconsistency may stem from the fact that subject representation uses BLIP-diffusion [27], which is a different base model than Stable Diffusion [46], as Stable Diffusion is fine-tuned. Additionally, in CUB, artistic prompts offer no improvement even when using Edge Guidance without subject representation, likely due to the dataset's heavy reliance on color as a primary discriminator between bird types, potentially disrupted by artistic prompts.

# 5 Limitations and Future Directions

**Limitations**. Although we demonstrated that SaSPA could generate images with high class fidelity through conditions such as edge maps and subject representation, it still remains dependent on the underlying generation models. For instance, we found that applying SaSPA to the CUB dataset at a higher resolution does not improve performance. Additionally, SaSPA relies on large language models (LLMs) to generate relevant and diverse prompts given the meta-class. While this is usually effective, it may not produce optimal prompts if the LLM lacks knowledge of the meta-class.

**Future Directions**. Several avenues exist to enhance the flexibility and performance of our method in future research. Firstly, we hope our work inspires the use of additional methods to condition the synthesis process beyond using real images, as we have shown to be effective. Another promising avenue is to apply SaSPA to additional tasks such as classification of common objects, object detection, and semantic segmentation. Additionally, maintaining temporal consistency in settings that use consecutive frames, such as autonomous driving, remains a significant challenge. Addressing this issue could expand the applicability of SaSPA to a broader range of use cases. Moreover, ongoing advancements in generative models are likely to bring further improvements to our pipeline. Finally, effectively generating and using synthetic data remains an active research area, and identifying optimal strategies for both the generation process and the training integration remains an important future direction.

# 6 Conclusion

We propose SaSPA, a generative augmentation method specifically designed for FGVC. Our method generates diverse, class-consistent synthetic images through conditioning on edge maps and subject representation. SaSPA consistently outperforms both traditional and recent generative data augmentation methods. It demonstrates superior performance across multiple settings, including multiple setups of the challenging and less-explored full dataset training, as well as in scenarios of contextual bias and few-shot classification. Limitations and future directions are discussed in Section 5.

# 7 Acknowledgments

This work was supported in part by the Israel Science Foundation (grant No. 1574/21).

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

# Supplementary Material

## Table of Contents

## A   Broader Impact

Generative data augmentation can benefit many fields by creating more robust models while protecting privacy. By reducing the reliance on real data, it addresses privacy concerns and lowers the costs and time needed for data collection and annotation, thereby enhancing the accessibility of advanced machine learning techniques. It is also important to note that synthetic data can inherit biases from the generative models, potentially leading to biased training outcomes.

## B   More Experiments

### B.1   Effect of Augmentation Ratio on Performance

In Figure 5, we evaluate various augmentation ratios (the probability of a real image being replaced by a synthetic one in a given mini-batch). We find that for most datasets, excluding CUB, the optimal

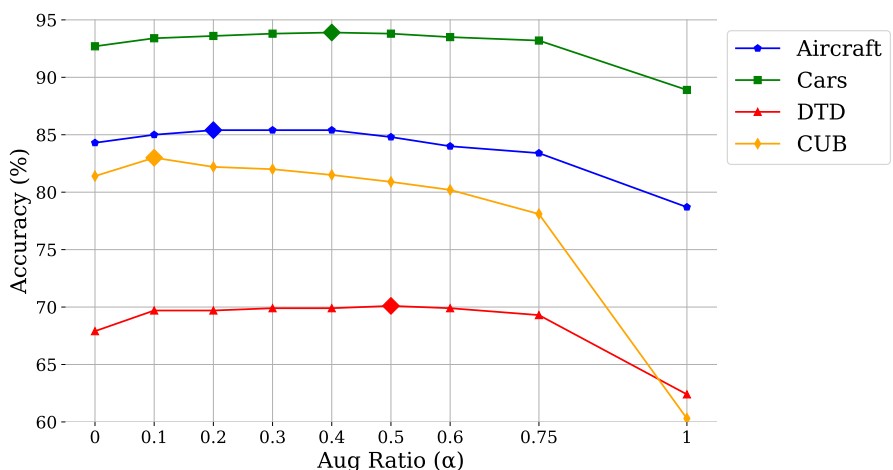

Figure 5: Line plots of Augmentation Ratio ($\alpha$) vs. validation accuracy for Aircraft, Cars, DTD, and CUB datasets.

Table 5: Effect of amount of real data used (as a fraction of the complete dataset) and $\alpha$ values on validation accuracy when augmenting with SaSPA

| Real Data | Aircraft | | | Cars | | | CUB | | |
|---|---|---|---|---|---|---|---|---|---|
| Frac. | Best Trad Aug | SaSPA ($\alpha$) | SaSPA ($\alpha_{high}$) | Best Trad Aug | SaSPA ($\alpha$) | SaSPA ($\alpha_{high}$) | Best Trad Aug | SaSPA ($\alpha$) | SaSPA ($\alpha_{high}$) |
| 0.1 | 26.9 | 40.8 | **41.0** | 29.3 | 50.5 | **51.6** | 32.7 | 38.4 | **41.4** |
| 0.3 | 59.7 | 69.8 | **70.0** | 70.8 | 83.9 | **84.3** | 61.3 | 66.2 | **68.3** |
| 0.5 | 73.5 | **78.7** | 77.9 | 84.7 | 89.3 | **89.5** | 72.0 | 74.8 | **76.0** |
| 0.75 | 80.6 | **82.9** | 82.5 | 90.7 | **92.6** | 92.3 | 77.6 | 80.3 | **80.7** |
| 1.0 | 84.3 | **85.4** | 84.0 | 92.7 | **93.9** | 93.6 | 81.4 | **83.0** | 82.0 |

range for $\alpha$ lies between 0.2 and 0.5, with marginal differences within this range. Consequently, we selected $\alpha = 0.4$ as the default augmentation ratio. However, for the CUB dataset, the default choice is $\alpha = 0.1$. Considering the relatively lower improvement on CUB in Table 1 and the underperformance on the *diff-mix* benchmark (Table 3), both of which are likely attributed to lower class-fidelity, it seems that lower class fidelity necessitates a lower augmentation ratio. A possible explanation is that higher augmentation ratios are more likely to introduce bias during training when class fidelity is lower.

## B.2 Effect of Augmentation Ratio on Performance with Different Amounts of Real Data

In Table 5, we examine the interaction between $\alpha$ and the percentage of real data used. We use two $\alpha$ values for each dataset: the default value used throughout the paper and a higher value ($\alpha_{high} = \alpha + 0.2$). We observe that for all amounts of real data, SaSPA achieves notable improvements with diminishing returns, similar to the trends observed in Section 4.3. An interesting pattern emerges: as the amount of real data decreases, the optimal value of $\alpha$ tends to increase. This trend is consistent across all datasets. For instance, in the Cars dataset, when all real data is used (Frac. 1.0), $\alpha = 0.6$ performs worse than $\alpha = 0.4$. However, for smaller percentages of real data (e.g., 10%, 30% or 50%), using $\alpha = 0.6$ yields better performance. This pattern is similarly observed in the Aircraft and CUB datasets, indicating that higher values of $\alpha$ are more beneficial when the amount of real data is limited.

## B.3 Comparing SaSPA with More Augmentation Methods

*diff-mix* [60] compared its method to more augmentation techniques. In this section, we present the complete results, including comparisons to those other methods.

Table 6: Comparison to concurrent work *diff-mix* [60]. Test accuracy on 3 different datasets. † indicates values taken from the diff-mix paper. *TI - Textual Inversion, DB - DreamBooth, ✗- No fine-tuning.*

| Aug. Method | FT Strategy | ResNet50@448 | | | ViT-B/16@384 | | |
|---|---|---|---|---|---|---|---|
| | | Aircraft | Car | CUB | Aircraft | Car | CUB |
| - | - | 89.09 | 94.54 | 86.64 | 83.50 | 94.21 | 89.37 |
| CutMix † | - | 89.44 | 94.73 | 87.23 | 83.50 | 94.83 | **90.52** |
| Mixup † | - | 89.41 | 94.49 | 86.68 | 84.31 | 94.98 | 90.32 |
| Real-filtering † | ✗ | 88.54 | 94.59 | 85.60 | 83.07 | 94.66 | 89.49 |
| Real-guidance † | ✗ | 89.07 | 94.55 | 86.71 | 83.17 | 94.65 | 89.54 |
| DA-fusion † | TI | 87.64 | 94.69 | 86.30 | 81.88 | 94.53 | 89.40 |
| Diff-Mix † | TI+DB | 90.25 | 95.12 | 87.16 | 84.33 | 95.09 | 90.05 |
| Diff-Mix + CutMix† | TI+DB | 90.01 | 95.21 | **87.56** | 85.12 | 95.26 | 90.35 |
| SaSPA (Ours) | ✗ | 90.59 | 95.29 | 86.92 | 85.48 | 95.12 | 89.70 |
| SaSPA (Ours) + CutMix | ✗ | **90.79** | **95.34** | 87.14 | **85.72** | **95.37** | 89.92 |

Table 7: Test performance of SaSPA combined with different traditional data augmentation methods.

| Type | Augmentation Method | Aircraft | CompCars | Cars | CUB | DTD |
|---|---|---|---|---|---|---|
| *Traditional* | No Aug | 81.4 | 67.0 | 91.8 | 81.5 | 68.5 |
| | CAL-Aug | 84.9 | 70.5 | 92.4 | 82.5 | 69.7 |
| | Best Trad Aug | - | 72.6 | 92.7 | - | - |
| *Ours* | SaSPA w/o Trad Aug | 84.1 | 74.1 | 92.8 | 81.7 | 69.7 |
| | SaSPA w/ CAL-Aug | **86.6** | 75.8 | **93.8** | **83.2** | **71.9** |
| | SaSPA w/ Best trad Aug | - | **76.2** | **93.8** | - | - |

**Experimental Setup.** As noted in Section 4.5, we use *diff-mix* training setup. This setup employs ResNet50 [20] with a resolution of $448^2$ and ViT-B/16 [14] with a resolution of $384^2$, both of which are higher than the $224^2$ resolution we use across the paper. We incorporate the integration of ControlNet and BLIP-diffusion for Cars and CUB datasets. We do not use BLIP-diffusion for the Aircraft dataset as it proved to be a better option, as evidenced in Table 4. The accuracies of *diff-mix* and other methods, as reported in Table 1 of the *diff-mix* paper [60], establish the benchmarks for our comparative analysis.

**Comparison Methods**. The compared methods, implemented by *diff-mix*, include (1) Real-Filtering (RF) and (2) Real-Guidance (RG), both proposed by He et al. [21]. RG is described in Appendix D.3, which they implement with a lower translation strength ($s = 0.1$). RF is a variation of Real-Guidance that generates images from scratch and filters out low-quality images by using CLIP [41] features from real samples to exclude synthetic images that resemble those from other classes. (3) DA-Fusion [54] solely fine-tunes the identifier using textual inversion [18] to personalize each sub-class and employs randomized strength strategy ($s \in \{0.25, 0.5, 0.75, 1.0\}$), and non-generative augmentation methods (4) CutMix [66] and (5) Mixup [67].

**Results.** Results show that *SaSPA* outperforms all methods across both architectures when evaluated on Aircraft and Cars, despite *diff-mix* using heavy fine-tuning. Continuing the discussion on CUB evaluation in Section 4.5, CutMix outperforms all methods when using the ViT-B/16 architecture, while *diff-mix* leads on ResNet50. Notably, *SaSPA* outperforms all other *generative* baselines on CUB except *diff-mix* using both architectures.

## B.4 Effect of Traditional Augmentations with SaSPA

Across our experiments, we combined SaSPA with the best traditional augmentation method, as described in Section 4.1. To test how SaSPA behaves without augmentation and whether it depends on the best traditional augmentation, we evaluated its interaction with CAL-Aug [44] (the default traditional augmentation used in CAL), the best traditional augmentation, and no traditional augmentation at all. Note that for 3 out of 5 datasets, CAL-Aug is the best traditional augmentation, so we

Table 8: Results on the test set of three FGVC datasets for ViT and ResNet101 architectures

| Aug Method | ViT | | | | ResNet101 | | |
|---|---|---|---|---|---|---|---|
| | Aircraft | Cars | DTD | ‖ | Aircraft | Cars | DTD |
| Best Trad Aug | 82.3 | 91.2 | 74.9 | ‖ | 85.5 | 93.2 | 69.3 |
| Real Guidance | 82.2 | 90.9 | 75.4 | ‖ | 85.1 | 93.0 | 70.3 |
| ALIA | 82.0 | 91.0 | 75.1 | ‖ | 85.0 | 92.9 | 69.4 |
| SaSPA | **86.1** | **91.5** | **76.3** | ‖ | **87.1** | **94.2** | **72.0** |

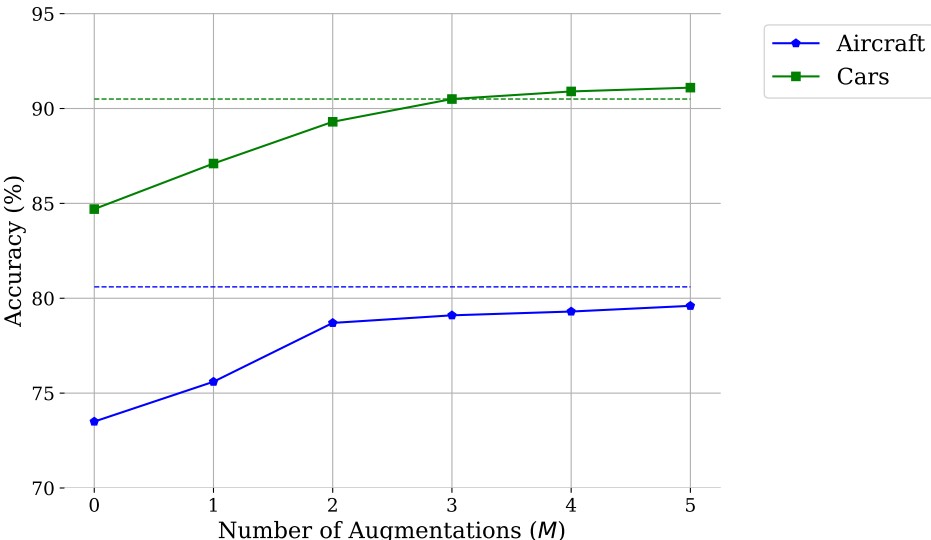

Figure 6: Effect of the number of *SaSPA* augmentations ($M$) on validation accuracy for Aircraft and Cars datasets. Horizontal lines represent the use of 75% real data without *SaSPA* augmentations.

did not provide separate results for SaSPA with the best traditional augmentation for these datasets (as they are the same). From the results in Table 7, we observe that using no traditional augmentation significantly under-performs compared to using CAL-Aug or the best traditional augmentation. Additionally, CAL-Aug proved to be a robust choice, yielding similar accuracy across all datasets, with only a slight decrease in performance for CompCars.

### B.5 Performance Across Network Architectures

Our primary experiments utilized ResNet50 as the backbone architecture for CAL. To further evaluate how deeper backbones or other network architectures might benefit from our augmentation method, we analyzed results across three FGVC datasets as detailed in Table 8. The performance of both ViT [14] and CAL with ResNet101 as a backbone is presented. Results indicate that both deeper networks, such as ResNet101, and the ViT architecture benefit from our augmentation method. Together with our comparison with *diff-mix* [60] using their training setup, this demonstrates that our method is robust across a variety of architectures and training setups.

### B.6 Effect of Scaling the Number of Augmentations (M)

In this study, we start with a base training set, utilizing 50% of the available real data. We examine the impact of varying the number of *SaSPA* augmentations, $M$, from 0 to 5 on the validation accuracy for Aircraft and Cars datasets. Additionally, we compare it to the effect of increasing the dataset size by adding an additional 25% of the real data without *SaSPA* augmentations.

Results presented in Figure 6 demonstrate a consistent increase in validation accuracy for both datasets as the number of augmentations increases. Notably, the Cars dataset shows robust performance

Table 9: **Additional datasets.** We report test accuracy on two additional FGVC datasets: Stanford Dogs and The Oxford-IIIT Pet Dataset. The highest values for each dataset are shown in **bold**.

| Augmentation Method | Pet | Dogs |
|---|---|---|
| CAL-Aug | 92.9 | 83.9 |
| Real Guidance | 92.9 | 83.5 |
| ALIA | 92.7 | 82.8 |
| SaSPA | **93.6** | **84.3** |

improvements, even surpassing the results achieved by adding 25% of real data when $M = 4$, indicating the effectiveness of the *SaSPA* augmentations in this context. For the Aircraft dataset, the accuracy nearly reaches the levels achieved by adding 25% more real data when $M = 5$.

## B.7 Extended Evaluation on Additional Datasets

In response to reviewer feedback, we expanded our evaluation to include two additional FGVC datasets: Stanford Dogs and the Oxford-IIIT Pet Dataset, to further assess the robustness of our proposed method, SaSPA. Stanford Dogs comprises 20,580 images from 120 dog breeds, while the Oxford-IIIT Pet Dataset includes 7,349 images from 37 breeds (25 dog and 12 cat breeds).

We compared with CAL-Aug, Real Guidance, and ALIA. Results, presented in Table 9, indicate that SaSPA improves performance on both datasets, thereby strengthening the findings regarding its efficacy as a generative augmentation method. Combined with earlier DTD and CUB datasets results, this evaluation confirms that SaSPA effectively handles non-rigid objects.

## B.8 Evaluating Different Prompt Strategies

Table 10: Comparison of prompt strategies across two FGVC datasets. The highest values are highlighted in **bold**, while the second highest are underlined.

| Prompt Strategy | Aircraft | Cars |
|---|---|---|
| Captions | 76.8 | 87.4 |
| LE | 78.3 | 87.9 |
| ALIA (GPT) | 78.2 | 88.1 |
| Ours (GPT) | 78.3 | 88.7 |
| Ours (GPT) + Art | **78.6** | **88.9** |

To assess the effectiveness of our proposed prompt generation, we evaluated various prompt strategies, and the results are detailed in Table 10. To accelerate the experimentation process, these experiments were conducted using only ControlNet with SD XL Turbo on 50% of the data. Five main strategies were compared: (1) **Captions**: Direct use of captions as prompts, leveraging BLIP-2 [28] for captioning, as demonstrated to be effective in prior work [25]. (2) **LE (Language Enhancement) [21]** and (3) **ALIA [15]** are described in Appendix D.3. (4–5) **Our Method** with and without appending artistic styles. The artistic style augmentation involves appending half of the prompts with the phrases ", a painting of <artist>", where <artist> refers to renowned artists such as van Gogh, Monet, or Picasso. This approach aims to diversify textures and colors, prompting an increase in the model's robustness.

The results show that our prompt generation method, either with or without incorporating artistic styles, consistently outperforms other approaches. Caption-based prompts yield the least effective performance, while the ALIA and LE methods fall somewhere in between.

## B.9 Will More Prompts Improve Performance?

To evaluate whether increasing the number of prompts would enhance our method's performance, we compared generating 200 prompts to generating 100 prompts using our method.

Table 11: Validation accuracy on the Aircraft dataset using 100 and 200 prompts generated by our method.

| Prompt Count | Accuracy |
|---|---|
| 100 | 78.9 |
| 200 | 79.0 |

The results in Table 11 show no significant difference when using 200 prompts, indicating that 100 prompts are sufficient.

## B.10 Assessing the Relevance of FID in Generative Data Augmentation

A common metric to evaluate the quality of a generative model is the Fréchet Inception Distance (FID) [22], a metric that measures the similarity between the distribution of generated images and real images. However, does it accurately measure how effective an *augmentation* method is?

Table 12: Combined FID and accuracy results for various generative augmentation methods across four FGVC datasets.

| Aug Method | Aircraft | | CompCars | | Cars | | CUB | |
|---|---|---|---|---|---|---|---|---|
| | FID | Acc. | FID | Acc. | FID | Acc. | FID | Acc. |
| Real Guidance | 3.39 | 84.8 | 8.74 | 73.1 | 9.08 | 92.9 | 5.93 | 82.8 |
| ALIA | 4.59 | 83.1 | 13.96 | 72.9 | 9.68 | 92.6 | 10.88 | 82.0 |
| SaSPA | 7.89 | **86.6** | 21.22 | **76.2** | 18.21 | **93.8** | 13.44 | **83.2** |

In Table 12, we report the FID values, calculated using augmentations alongside their respective real datasets, as well as the corresponding accuracy achieved with each augmentation method. We observe that generative baselines such as Real Guidance and ALIA achieve lower FID scores, which suggest a higher similarity to the real data distribution. We suspect that this is the result of generating images that closely mimic the original dataset. In contrast, our method, SaSPA, is designed to create diverse augmentations that substantially differ from the real images, leading to higher FID scores. Despite these higher FID values, as shown in Table 12, SaSPA demonstrates superior performance enhancements in accuracy across datasets. This highlights the importance of evaluating generative augmentation methods not only based on realism and similarity to real images, as measured by FID, but primarily on their actual impact on model performance. In the next section, we further provide an alternative metric for generative data augmentation.

## B.11 Evaluating Augmentation Diversity with LPIPS

LPIPS [69] measures the perceptual difference between two images. By calculating the average LPIPS distance between original images and their respective augmentations, we can quantify the *diversity* introduced by an augmentation method. We argue that this metric, combined with qualitative evidence of class fidelity, provides a robust measure for evaluating generative data augmentation. Note that this metric will apply only for augmentations that are derived from real images. Generation from scratch will require a different metric, probably a *dataset-level* diversity metric.

Table 13: Combined diversity score and accuracy results for various generative augmentation methods across five FGVC datasets.

| Aug Method | Aircraft | | CompCars | | Cars | | CUB | | DTD | |
|---|---|---|---|---|---|---|---|---|---|---|
| | Diversity | Acc. | Diversity | Acc. | Diversity | Acc. | Diversity | Acc. | Diversity | Acc. |
| Real Guidance | 0.11 | 84.8 | 0.10 | 73.1 | 0.10 | 92.9 | 0.15 | 82.8 | 0.18 | 68.5 |
| ALIA | 0.24 | 83.1 | 0.29 | 72.9 | 0.26 | 92.6 | 0.33 | 82.0 | 0.37 | 69.1 |
| SaSPA | **0.55** | **86.6** | **0.53** | **76.2** | **0.57** | **93.8** | **0.66** | **83.2** | **0.58** | **71.9** |

Table 13 demonstrates that SaSPA achieves significantly higher LPIPS scores compared to Real Guidance (RG) and ALIA, indicating that SaSPA introduces much greater diversity in the generated augmentations. This substantial increase in diversity is crucial for enhancing model robustness and performance [31]. Qualitative evidence can be found in Figure 7.

## B.12  Investigating the Potential of Newer Base Models

Table 14: Validation accuracy of our method with different base models. Generations do not include BLIP-diffusion.

| Base Model | Edge Guidance | Aircraft | CompCars | Cars | CUB |
|---|---|---|---|---|---|
| Best Trad Aug | - | 84.3 | 62.5 | 92.7 | 81.4 |
| SD v1.5 |  | 83.3 | 63.1 | 92.8 | 82.1 |
|  | ✓ | 86.2 | 64.7 | 93.8 | 81.8 |
| SD XL Turbo |  | 83.5 | 62.8 | 93.5 | 82.2 |
|  | ✓ | 86.4 | 65.0 | 93.7 | 82.3 |
| SD XL |  | 83.8 | 62.7 | 93.4 | 82.6 |
|  | ✓ | 86.7 | 64.6 | 93.7 | 82.3 |

Recently, text-to-image diffusion models have made incredible progress, particularly those based on Stable Diffusion (SD) [46]. Notable advancements include SD XL [40] and SD XL Turbo [51]. In this section, we aim to explore the compatibility of Edge Guidance with other base models. Unfortunately, as these models are relatively new, BLIP-diffusion [27] has not yet released versions built upon them, preventing us from utilizing subject representation. However, as shown in Table 4, our full pipeline without subject representation still achieves impressive results. Additionally, Table 4 indicates that when subject representation is not used, it is slightly beneficial for most datasets, except CUB, to append half the prompts with artistic styles. Therefore, we adopt this strategy. In Table 14, we experiment with SD v1.5, SD XL, and SD XL Turbo. Note that ControlNet versions for SD XL and SD XL Turbo are still experimental, and will require reevaluation as the models mature.

The results indicate that integrating Edge Guidance generally has a positive impact across base models, except on the CUB dataset, which aligns with our earlier findings in Table 4. Additionally, SD XL and SD XL Turbo typically outperform SD v1.5, suggesting that more advanced base models may lead to further improvements in performance.

## B.13  Does Stopping Augmentation at Early Epochs Help?

Table 15: Impact of stopping *SaSPA* augmentation at different training epochs on validation accuracy of the Aircraft [30] dataset.

| Epoch Stop | Accuracy |
|---|---|
| 0 (No Aug) | 73.5 |
| 20 | 75.8 |
| 40 | 77.2 |
| 60 | 77.1 |
| 80 | 77.3 |
| 100 | 77.4 |
| 120 | 77.7 |
| 140 (Full Aug) | 78.7 |

A common technique when training with synthetic data is to first train using the synthetic data, then fine-tune on the real data [56, 38, 17]. Inspired by this, we investigate stopping the data augmentation at earlier training epochs. Results are presented in Table 15. For these ablation experiments on evaluating early augmentation stoppage, we used 50% of the real Aircraft dataset for faster experimentation. We find no benefit from early augmentation stoppage. There is a downward trend in accuracy when stopping early, with the worst results at the earliest epoch stopped (20 out

of 140). We believe that the high diversity introduced by our augmentations reduces over-fitting, mitigating the need for an explicit domain adaptation strategy. As a result, the model continues to benefit from the augmented data throughout the training process.

### B.14 Performance at Higher Resolutions

Table 16: **Higher resolution results.** Comparison of our method (SaSPA) with the best augmentation method per dataset. All results are using 448x448 resolution, and reported on the test set of each dataset.

| Method | CompCars | DTD | CUB |
|---|---|---|---|
| Best Aug | 75.1 | 69.6 | **86.7** |
| SaSPA | **77.6** | **72.0** | **86.7** |

Additional experiments were conducted using a 448x448 resolution on the CompCars, DTD, and CUB datasets, employing SaSPA and the best augmentation methods identified in Table 1. The experiments, replicated with two different seeds, are detailed in Table 16.

Combined with our earlier diff-mix comparisons at both 448x448 and 384x384 resolutions (Table 3), these results complete our high-resolution evaluation across all datasets. Notably, we observed consistent performance improvements in all datasets except CUB. We hypothesize that CUB's fine-grained details such as feather patterns and colors present significant challenges at higher resolutions, impacting the efficacy of generative methods. In conclusion, SaSPA demonstrates promising results for most datasets, affirming its overall benefits.

### B.15 Choice of Conditioning Type

Table 17: Validation accuracy on the Aircraft dataset using different conditioning types of ControlNet.

| Condition Type | Accuracy |
|---|---|
| Canny Edges | 78.9 |
| HED Edges | 78.6 |

Using Canny edge maps [6] as a condition has proven effective for generating images with high diversity and class fidelity. Here, we experiment with a different kind of edges: Holistically-Nested Edge Detection (HED) edges [63]. Canny edges are more focused on detecting the intensity gradients of the image, often capturing finer details, whereas HED edges provide a more structured representation by capturing object boundaries in a holistic manner. We experiment with both types in Table 17, using the default generation parameters without using BLIP-diffusion, and on 50% of the data for faster experimentation. Using Canny edges resulted in slightly higher validation accuracy on the Aircraft dataset.

## C   Dataset details

We provide the number of samples for each dataset split used in our experiments in Table 18. Additionally, we include the number of images for each background class (sky, grass, road) used to create the contextually biased training set, as shown in Table 19. We utilize the dataset test split for the reported test. For datasets lacking a validation split (Cars, CUB, CompCars), we generate one by using 33% of the training set. Note that in the *diff-mix* training setup (Section 4.5), the training datasets for CUB and Cars consist of the original splits, as they did not use a separate validation split.

## D   Implementation details

### D.1   Experimental Setup

Unless stated otherwise, the following experimental setup applies to all experiments in the paper.

Table 18: Dataset Split Sizes.

| Dataset | Training | Validation | Testing |
|---|---|---|---|
| Aircraft | 3,334 | 3,333 | 3,333 |
| CompCars | 3,733 | 1,838 | 4,683 |
| Cars | 5,457 | 2,687 | 8,041 |
| CUB | 4,016 | 1,978 | 5,794 |
| DTD | 1,880 | 1,880 | 1,880 |
| Airbus VS Boeing | 409 | 358 | 707 |

Table 19: Dataset Statistics for Contextually Biased Planes

| | Airbus | | | Boeing | | |
|---|---|---|---|---|---|---|
| | sky | grass | road | sky | grass | road |
| Train | 98 | 0 | 70 | 129 | 112 | 0 |
| Val | 90 | 21 | 21 | 137 | 45 | 44 |
| Test | 175 | 51 | 51 | 222 | 104 | 104 |

**Data Generation.** All generative methods use the Diffusers library [57]. We employ BLIP-diffusion [27] and ControlNet [68] for *SaSPA*. Besides the prompt, an edge map for ControlNet, and a reference image for BLIP-diffusion as inputs for our generation, BLIP-diffusion requires source subject text and target subject text as inputs. We simply use the *meta-class* (e.g., "Airplane" for Aircraft dataset, "Bird" for CUB) of the dataset for both source and target subject texts. For all other diffusion-based augmentation methods, we use Stable Diffusion v1.5 [46]. For all diffusion-based models, including *SaSPA*, we use the DDIM sampler [53] with 30 inference steps and a guidance scale of 7.5. Images are resized to ensure the shortest side is 512 pixels before processing with Img2Img or ControlNet. We set the ControlNet conditioning scale to 0.75. For text-to-image generation, images are generated at a resolution of 512x512. We generate $M = 2$ augmentations per original image for each experiment, and we use augmentation ratio $\alpha = 0.4$ for all datasets except CUB [58], for which we use $\alpha = 0.1$ as evidenced to be better in Figure 5. We use $k = 10$ in the top-k Confidence filtering. We use four NVIDIA GeForce RTX 3090 GPUs for image generation and training.

**Training.** We follow the implementation strategy outlined in the CAL study [44], tailored for FGVC. We use ResNet50 [20] as the primary architecture within the CAL framework unless specified otherwise. Each dataset is fine-tuned using pre-trained ImageNet [12] weights. Optimization is performed with an SGD optimizer, with a momentum of 0.9 and a weight decay of $10^{-5}$, over 140 epochs. We adjust the learning rate and batch size during hyper-parameter tuning to achieve the highest validation accuracy. Training images are resized to 224x224 pixels. Results are averaged across three seeds. Specific values of hyper-parameters are in Table 20.

**Specifics on DTD [9] dataset.** The DTD dataset is a collection of images categorized by various textures, such as Marbled, Waffled, and Banded. We found that this dataset differs from other fine-grained datasets as it is not fine-grained at the same level. Classes like "Marbled" and "Waffled" have significant differences from each other. Therefore, feeding the *meta-class* ("Texture") to an LLM will provide prompts that are not suitable for all *sub-classes* in the dataset. Hence, we did not use our prompt generation method. This could be addressed in the future by feeding the LLM with each sub-class. Instead, we simply used image captions.

**Hyper-parameters** To select the hyper-parameters for each dataset, we train CAL [44] with learning rates of [0.00001, 0.0001, 0.001, 0.01, 0.1] and batch size [4, 8, 16, 32], selecting the configuration that results in the highest validation accuracy. These parameters, shown in Table 20, are then used across all methods.

## D.2 Compute Requirements

In this section, we outline the computational resources required for our primary experiments. We utilize four NVIDIA GeForce RTX 3090 GPUs for image generation and training purposes, but we report running times for a single GPU. Training with ResNet50 necessitates up to 5.5 GB of GPU RAM. The duration of our experiments varies depending on the dataset, with the longest running being approximately three hours. As no fine-tuning is performed, our generation process includes

Table 20: Hyperparameters

| Dataset | Learning Rate | Batch Size | Weight Decay | Epochs | Optimizer | Momentum |
|---|---|---|---|---|---|---|
| Aircraft [30] | 0.001 | 4 | $10^{-5}$ | 140 | SGD | 0.9 |
| CompCars [64] | 0.001 | 8 | $10^{-5}$ | 140 | SGD | 0.9 |
| Cars [24] | 0.001 | 8 | $10^{-5}$ | 140 | SGD | 0.9 |
| CUB [58] | 0.001 | 16 | $10^{-5}$ | 140 | SGD | 0.9 |
| DTD [9] | 0.001 | 16 | $10^{-5}$ | 140 | SGD | 0.9 |
| Airbus vs. Boeing [15] | 0.001 | 4 | $10^{-5}$ | 140 | SGD | 0.9 |

only I/O and a forward pass through the generation model. We report here only the generation times and do not include I/O times, as these can vary heavily based on system configuration and server load. For image augmentation using ControlNet with BLIP-diffusion as the base model, generating each image takes 2.96 seconds and requires up to 10 GB of GPU memory. Therefore, creating two augmentations for the Aircraft dataset's training set would take approximately five and a half hours. When switching to SD XL Turbo as the base model, with two inference steps (the default for this base model), the augmentation time is reduced to 0.52 seconds, and the GPU memory requirement increases to up to 16 GB. In this configuration, generating two augmentations for the Aircraft dataset's training set would take less than one hour.

### D.3   More details on Generative Baselines

In this section, we provide additional details on the generative baselines we compared against.

**Real-Guidance (RG)**: This method achieves impressive few-shot classification performance. For prompt generation, an off-the-shelf word-to-sentence T5 model, pre-trained on the "Colossal Clean Crawled Corpus" [42] and fine-tuned on the CommonGen dataset [29], is utilized to diversify language prompts. The model is used in order to generate a total of 200 prompts based on the *meta-class*. For image generation, SDEdit with a low translation strength ($s = 0.15$) is used. Filtering is performed using CLIP filtering, which is described in Section 3.4.

**ALIA [15]**: This method showed impressive results in addressing contextual bias and domain generalization. For prompt generation, GPT-4 [1] is employed to summarize image captions of the training dataset into a concise list of fewer than 10 domains, which are then used inside prompts. Image generation is carried out using either SDEdit with medium strength (around 0.5) or InstructPix2Pix [5]. For filtering, they use a confidence-based filtering approach where a model $f$ is trained, and a confidence threshold $t_y$ for each class $y$ is established by averaging the softmax scores of the correct labels from the training set. An edited image $x'$ with a predicted label $\hat{y}$ is excluded if confidence$(f(x'), \hat{y}) \geq t_{\hat{y}}$. This thresholding ensures that images for which the predicted label $\hat{y}$ matches the true label $y$ are removed due to redundancy. Additionally, images where $\hat{y} \neq y$ with high confidence are also filtered out because they likely represent a significant alteration, making them resemble another class more closely. *For our pipeline*, we observe that this approach tends to overly filter augmentations where $\hat{y} = y$, as augmentations that could be redundant due to similarity to the original real image do not occur in our method, as we do not use real images as guidance for augmentation.

## E   More Methodology details

### E.1   Prompt Generation via GPT-4

In this section, we provide more details on how we used GPT-4 to create prompts. After identifying the *meta-class* of the FGVC dataset, we input it into the following instruction:

"Generate 100 prompts for the class [meta-class] to use in a text-to-image model. Each prompt should:

- Include the word [meta-class] to ensure the image focuses on this object.

- Ensure diversity in each prompt by varying environmental settings, such as weather and time of day. You can include subtle enhancements like vegetation or small objects to add depth to the scene, ensuring these elements do not narrowly define the [meta-class] beyond its broad classification.

• The prompts should meet the specified quantity requirement."

No quality control is used over the generated prompts.

## E.2 Filtering Strategies

This section elaborates on our filtering mechanisms that remove lower-quality augmentations that do not correctly represent the *sub-class* or the *meta-class*. Additionally, we compare the effectiveness of alternative filtering methods in Table 21.

**Predictive Confidence Filtering** utilizes the baseline model's confidence to filter out augmentations whose true label does not rank within the model's top-k predictions ($k = 10$). This baseline model is selected based on optimal performance outcomes from a hyperparameter sweep, as described in Appendix D.1. The choice of $k$ can affect the results: using too low of a $k$ can result in excessive filtering, limiting augmentations to those the baseline model already handles well, whereas too high of a $k$ results in insufficient filtering, allowing low-quality augmentations to pass through. Therefore, we ablate on $k$ as well to find the optimal value. We show a visualization of this filtering method in Figure 8.

We also evaluate other filtering methods, including **CLIP filtering** [21], **Semantic Filtering**, and **ALIA confidence filtering** [15], as described in Section 3.4 and Appendix D.3.

Note that for certain datasets, such as Cars [24] and Aircraft [30], the augmentations remain so consistent that only minimal filtering is required. For instance, out of the total augmentations produced for the Cars dataset and using our filtering method, only 0.1% were filtered out. However, this percentage is more significant for other datasets like CompCars [64], where 4.5% of augmentations were filtered.

Table 21: Performance of different filtering methods on the CompCars validation dataset, highlighting the effectiveness of combined and individual strategies.

| Prompt Strategy | Accuracy |
| --- | --- |
| No filter | 49.4 |
| CLIP Filtering | 48.1 |
| Semantic Filtering | 49.6 |
| ALIA Confidence Filtering | 49.6 |
| ALIA Confidence Filtering + Semantic Filtering | 49.8 |
| Top-1 Confidence Filtering | 47.4 |
| Top-5 Confidence Filtering | 49.6 |
| Top-10 Confidence Filtering | 49.8 |
| Top-20 Confidence Filtering | 49.4 |
| Top-10 Confidence filtering + Semantic filtering | **50.1** |

Results from employing the various filters are presented in Table 21. Note that for faster experimentation, we used 50% of the data (hence the low accuracy). Observations include: (1) CLIP filtering leads to poorer performance than using no filter, likely because CLIP struggles with fine-grained concepts such as specific car model tail lights. (2) Our confidence filtering method achieves the best results at $k = 10$. (3) Combining our confidence filtering with semantic filtering surpasses all other methods.

## F  More Visualizations

### F.1  Qualitative Comparison with Generative Augmentation Methods

Example augmentations of Real Guidance, ALIA, and our method are visualized in Figure 7.

### F.2  Confidence Filtering Visualization

Examples of augmentations that were and were not filtered for three FGVC datasets are in Figure 8.

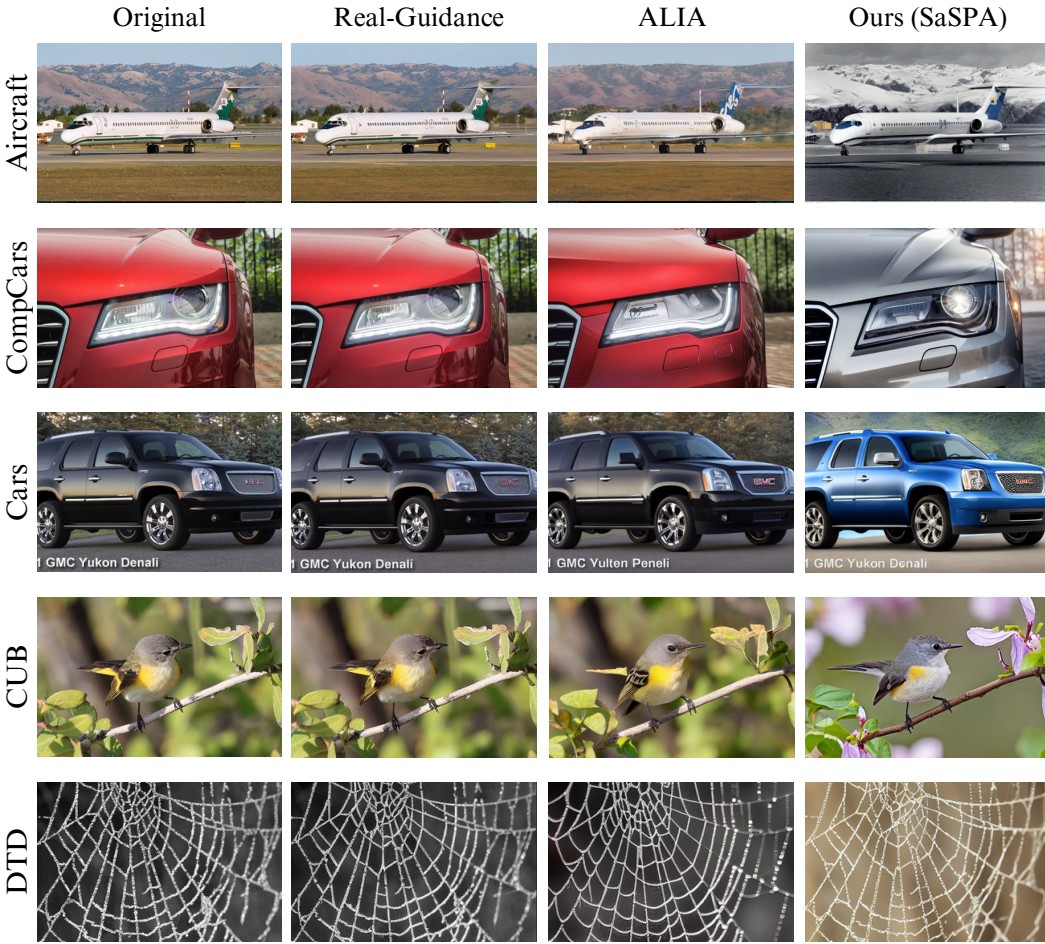

Figure 7: Qualitative results of different generative augmentation methods: Real-Guidance, ALIA, and SaSPA on five FGVC datasets. Real Guidance produces very subtle variations from the original image due to the low translation strength they used. ALIA generates visible variations, but they are considerably less diverse compared to the augmentations produced by SaSPA.

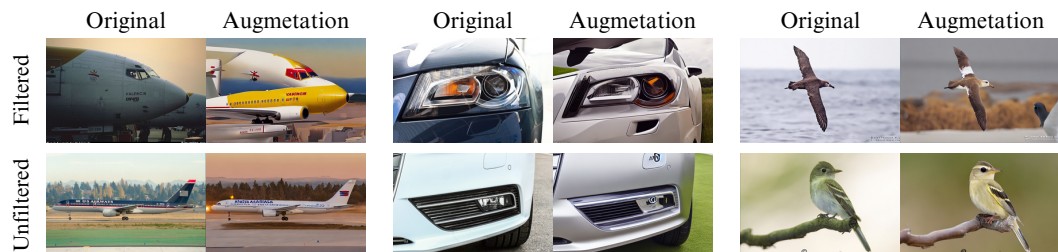

Figure 8: Randomly selected augmentations of SaSPA that were and were not filtered for Aircraft, CompCars, and CUB.

