# OpenReview forum: "Advancing Fine-Grained Classification by Structure and Subject Preserving Augmentation"
_NeurIPS.cc/2024/Conference — NeurIPS 2024 poster_

### Official Review · Reviewer_Z7K4 · 2024-07-09

**Soundness:** 3
**Presentation:** 3
**Contribution:** 3
**Rating:** 7
**Confidence:** 4

**Summary:**

The paper proposes SaSPA, which uses several large, pre-trained generative, language, and vision-language models to produce high-quality synthetic images. Specifically they focus on images for FGVC, where the high inter-class similarities make it challenging to synthesize data that is not only diverse (capturing the intra-class variability) but also faithful to the intended class label. They combined GPT-4 with BLIP ControlNet diffusion, with edge map conditioning and post-generation filtering (via top-10 ranking with CLIP) to achieve performance gains compared to baselines, with only 2 augmentations per image (notably less than the 10x once thought to be a rule of thumb for training with synthetic data).

**Strengths:**

[S1] SaSPA seems well-motivated, rather than haphazardly throwing large language and generative models at the task, the paper presents a carefully crafted pipeline that addresses the fidelity issue with both strong conditions (such as the edge maps) and also filtering.

[S2] The paper is mostly well-presented, easy to follow, with motivation, method, and results that flow well both narratively and logically.

[S3] The experiments are, for the most part, very thorough. (that's not to say nothing is missing, see W1)

This does not matter so much, but

[S4] The differences between the proposed method and diff-mix are well-articulated, and SaSPA has some clear advantages to mitigate its disadvantages (such as not requiring additional training of the generative pipeline).

**Weaknesses:**

The results do not deliver on the promise of the paper's motivation. Principally,

[W1] It is unclear if this approach can contribute to some state-of-the-art result. In some areas (e.g. text-to-image generation itself), asking SOTA of a research people might represent an unreasonable burden. However, in an area like FGVC, such evaluations are both practical and necessary. The authors should show that the performance gains on the ResNet50 (Table 1) don't vanish when applied to stronger methods (and I don't really count ResNet101 here, it's barely better than R50), particularly for CUB.

Speaking of CUB, it seems from the teaser that this method may struggle to capture the most fine-grained differences, such as when it recolors the engine. Perhaps this is why the method does not fare so well with the birds dataset, or in other words,

[W2] The method's inferiority to CutMix on CUB for 448x448 images is very concerning (Table 3). The higher resolution is the standard for FGVC, and a method that only works well for low resolution is quite limited in impact. Additionally, it calls into question whether the method is good mainly for images of rigid objects, or if it could work well for other datasets of living things (Stanford Dogs, Fungi, NABirds).

On a more minor note,

[W3] Some information seems misplaced. Image resolution is critical context in FGVC, especially if the paper uses smaller-than-normal images. The comparison in Table 3 should also compare M for diff-mix and the SaSPA.

and

[W4] Using CLIP for the filtering step seems strange. A model trained on the dataset (or even an ensemble) would likely filter the data much more reliably. Would this not be a much stronger approach?

**Questions:**

My weaknesses correspond to questions.

[W1] - If this data augmentation is applied to a SOTA FGVC model, is the impact orthogonal, e.g., do I get a new SOTA? Or is this just another way to get the model to learn the same information as the existing tricks?

[W2] - Does this method work well beyond Cars/Aircraft/DTD? Will it break for other datasets, similar to how it seems to break for CUB?

[W4] - How was CLIP selected for the filtering step?

I am currently leaning reject, mainly due to W1 and W2. For clarity's sake, I have similar concerns about diff-mix and would not necessarily have voted to accept, had I reviewed that paper. I say this just because W1 applies to diff-mix as well, and replying to my review by pointing this out will not assuage my concern.

**Limitations:**

Yes.

---

> ### Author Rebuttal · Authors · 2024-08-07
>
> Thank you for your detailed review and insightful feedback. We appreciate your recognition of SaSPA's well-motivated and carefully crafted pipeline, as well as your positive comments on the presentation and thoroughness of our experiments. Your constructive points will help us improve our work further.
>
> **“to achieve performance gains compared to baselines, with only 2 augmentations per image (notably less than the 10x once thought to be a rule of thumb for training with synthetic data).”**
> Thank you for highlighting this! We have now included this point in the paper.
>
> **“[W1] It is unclear if this approach can contribute to some state-of-the-art result. In some areas (e.g. text-to-image generation itself), asking SOTA of a research people might represent an unreasonable burden. However, in an area like FGVC, such evaluations are both practical and necessary. The authors should show that the performance gains on the ResNet50 (Table 1) don't vanish when applied to stronger methods (and I don't really count ResNet101 here, it's barely better than R50), particularly for CUB.”**
> - We'd like to clarify that our baseline architecture is not simply a standard ResNet, but rather the CAL [1] architecture, which is tailored for FGVC and was SoTA at the time of its publication. ResNet is used as a backbone in this architecture.
> - We have updated the paper to make this distinction clearer, preventing any potential confusion for future readers.
> - While CAL may not be the current SoTA for some datasets, it remains a strong and relevant baseline for FGVC tasks. For instance, MetaFormer [2], suggested by reviewer JRLS, shows mixed results when compared to CAL (MetaFormer is better than CAL on CUB but performs worse on Cars, with similar results on aircraft).
> - Nevertheless, we acknowledge the value of evaluating our method on more architectures, and we will have results for a more recent architecture for the camera-ready version of our paper.
>
> **“The method's inferiority to CutMix on CUB for 448x448 images is very concerning (Table 3). The higher resolution is the standard for FGVC, and a method that only works well for low resolution is quite limited in impact.”**
> Please refer to the global rebuttal, we have added experiments on high-resolution images.
>
> **“Additionally, it calls into question whether the method is good mainly for images of rigid objects, or if it could work well for other datasets of living things (Stanford Dogs, Fungi, NABirds).”**
> Please refer to the global rebuttal, we have added experiments on two new datasets: Dogs and Pet.
>
> **“[W3] Some information seems misplaced. Image resolution is critical context in FGVC, especially if the paper uses smaller-than-normal images. The comparison in Table 3 should also compare M for diff-mix and the SaSPA.”**
> - Regarding image resolution, we have updated our manuscript to make the resolution clearer.
> - Please also refer to the global rebuttal, we have added results for higher-resolution images as well.
> - Regarding the comparison in Table 3, generally speaking, we have two training schemes in our paper: (1) our training scheme and (2) the diff-mix training scheme. We opted to use the diff-mix training scheme when comparing to it because this provides a stronger comparison, favoring diff-mix as they trained and validated using their own training scheme and parameters. Hence, we cited diff-mix results from their paper rather than training using our pipeline, including for the few-shot scenarios.
> - We have used the same M for SaSPA as diff-mix used to ensure an apples-to-apples comparison. We updated the manuscript with this information.
> - It is important to note that diff-mix is a **concurrent paper**. According to the official guidelines, authors are not expected to compare to such works. Nevertheless, we chose to add this comparison for completeness. The methods are quite different, and we believe both are valuable for the research community. Unlike Diff-Mix, which uses fine-tuning for its generative model, our method does not rely on such heavy fine-tuning. We have made it clearer in the paper that diff-mix is a concurrent work, as this point may have been missed.
>
> **“[W4] Using CLIP for the filtering step seems strange. A model trained on the dataset (or even an ensemble) would likely filter the data much more reliably. Would this not be a much stronger approach?”**
> - Our methods uses 2 filtering methods:
>   - (1) Semantic filtering, proposed by a recent work ALIA [3] to alleviate meta-class misrepresentation. Using **CLIP,** this process evaluates the relevance of generated images to the specific task at hand. For example, in a car dataset, each generated image is assessed against a variety of prompts such as “a photo of a car”, “a photo of an object”, “a photo of a scene”, “a photo”, and “a black photo”. Images that CLIP does not recognize as “a photo of a car” are excluded to ensure that the augmented dataset closely aligns with the target domain.
> - (2) Our top-10 filtering, which uses a **model trained on the dataset**, as you suggested. We made it clearer in the paper.
> - Note: Your intuition is correct! As we show in the paper (appendix F.2 Filtering Strategies), top-10 filtering is more important than semantic filtering, as evident in Table 19.
>
>
>
>
> [1] Rao, Yongming, et al. "Counterfactual attention learning for fine-grained visual categorization and re-identification." Proceedings of the IEEE/CVF international conference on computer vision. 2021.
> [2] Diao, Qishuai, et al. "Metaformer: A unified meta framework for fine-grained recognition." arXiv preprint arXiv:2203.02751 (2022).
> [3] Dunlap, Lisa, et al. "Diversify your vision datasets with automatic diffusion-based augmentation." Advances in neural information processing systems 36 (2023): 79024-79034.

---

> > ### Comment · Reviewer_Z7K4 · 2024-08-07
> > **Concerns Resolved**
> >
> > All my weaknesses have been properly addressed. My apologies for the confusion on your Predictive Confidence Filtering (and corresponding F.2).
> >
> > As far as diff-mix goes, I realize now that I wasn't clear in my review, but I realize that it is concurrent work (that's why I mentioned it as minor, on both strengths and weaknesses). Presenting it isn't mandatory, but given that it has presented, preferably it would be presented properly (and it seems the main issue was corrected). As far as my evaluation of this work, and my rating, the comparison to diff-mix is only a positive, insofar as it reflects this work's thoroughness.
> >
> > I would suggest potentially adding a clearly marked limitations section to the final manuscript to address lingering issues such as less satisfying performance on high resolution CUB.

---

> > > ### Author Response · Authors · 2024-08-08
> > >
> > > We are happy that all your concerns are resolved. We will incorporate the clarifications, and add a clearly marked limitation section as you suggested.

---

### Official Review · Reviewer_JRLS · 2024-07-12

**Soundness:** 3
**Presentation:** 4
**Contribution:** 3
**Rating:** 7
**Confidence:** 4

**Summary:**

The paper proposes a data-augmentation technique tailored to fine-grained image classification.
The goal is to increase class fidelity while maintaining high variance in the images, something current diffusion-based data augmentation techniques are struggling with, especially in the fine-grained domain.

The main idea is to preserve both the scene-structure (by conditioning on edges with controlnet) and subjects (by using blip diffusion). Rather than augmenting a single sample, the edge map, class subject reference image and prompt are sampled independently. An image filtering strategy is proposed to increase the data quality. The method doesn’t require any finetuning of the generative model.

The method is shown to outperform baselines on 5 datasets and different scenarios such as full dataset training and few-shot classification are explored. Better generalization to unseen backgrounds compared to classifiers trained on the data augmentation baselines is also shown.

**Strengths:**

1. The method builds on existing techniques, combining them in a novel way, which may lead to followup works
2. The method is well-contextualized within related work
3. The method is simple (easy to udnerstand and implement), novel and addresses an important problem of training deep learning models on domains with limited amount of training data, in particular for fine-grained visual classification
4. The paper is well written, easy to follow, with well-designed figures explaining the methodology
5. Extensive evaluation including many ablation expeirments justifying the design choices
6. Experiments are well described with all the necessary details such as information on hyper-parameter tuning, code will also be released for reproducibility

**Weaknesses:**

1. The baseline classifier architecture is a standard image classifier. Meanwhile, there are architectures tailored to the fine-grained classification domain such as the MetaFormer [1, 2], it would be preferrable to also evaluate on state-of-the-art classifiers from the FGVC domain.
2. It is not clear how the method would perform on a dataset of objects from a less common domain (not so well represented in the training data of the diffusion models)
See Questions for more details.

**Questions:**

## General remarks
1. Code release is only mentioned in section 3.5, which is easy to miss, I would place it in abstract/introduction.
2. Even though it is proposed for FGVC, the method is in principle applicable elsewhere - have you tried applying it to standard image classification datasets? Is there a reason why you think the method would not work well?

## Add weakness 2
3. The evaluation datasets consist of very common objects/animals which are well-represented in the training data of diffusion models such as cars and birds. It would help understanding the strenghts and weaknesses of the method to see its performance on a dataset with less common meta-classes, for example the iNaturalist.


**References**:

[1] Diao, Qishuai, et al. "Metaformer: A unified meta framework for fine-grained recognition." arXiv preprint arXiv:2203.02751 (2022).

[2] He, Ju, et al. "Transfg: A transformer architecture for fine-grained recognition." Proceedings of the AAAI conference on artificial intelligence. Vol. 36. No. 1. 2022.

**Limitations:**

Limitations are adequately addressed in the appendix.

---

> ### Author Rebuttal · Authors · 2024-08-06
>
> We appreciate your recognition of our method's novelty and effectiveness in addressing training with limited data. We're glad you found the paper well-written and the evaluations helpful. Our code has already been released for reproducibility.
>
> **“The baseline classifier architecture is a standard image classifier. Meanwhile, there are architectures tailored to the fine-grained classification domain such as the MetaFormer [1, 2], it would be preferable to also evaluate on state-of-the-art classifiers from the FGVC domain.”**
> - We'd like to clarify that our baseline architecture is not simply a standard ResNet, but rather the CAL [1] architecture, which is tailored for FGVC and was SoTA at the time of its publication. ResNet is used as a backbone in this architecture.
> - We have updated the paper to make this distinction clearer, preventing any potential confusion for future readers.
> - While CAL may not be the current SOTA for some datasets, it remains a strong and relevant baseline for FGVC tasks. For instance, MetaFormer shows mixed results when compared to CAL (MetaFormer is better than CAL on CUB but performs worse on Cars, with similar results on aircraft).
> - Nevertheless, we acknowledge the value of evaluating our method on more architectures like MetaFormer, and we will have results for another architecture such as the MetaFormer for the camera-ready version of our paper.
>
>
>
> **“Code release is only mentioned in section 3.5, which is easy to miss, I would place it in abstract/introduction.”**.
>
> Thank you for the suggestion. We have moved the code release information from section 3.5 to the abstract/introduction to increase visibility.
>
> **“Even though it is proposed for FGVC, the method is in principle applicable elsewhere - have you tried applying it to standard image classification datasets? Is there a reason why you think the method would not work well?”**
> - Thank you for considering the broader applicability of our method. There is no inherent reason our method should not work well on standard image classification datasets.
> - We focused on FGVC after identifying a gap in the existing literature concerning the application of diffusion models as generative augmentation within this domain. FGVC poses unique challenges and opportunities due to the subtle distinctions between classes, the struggles of current generative models to create images with accurate subclass fidelity, and the scarcity of data.
> - We believe our method has the potential to improve standard image classification as well, given its strong performance in generating diverse and accurate images for fine-grained categories. While our current focus was on FGVC, we are excited about the possibility of extending our approach to more general datasets in future work. Hence, we have updated our future work section.
>
> **“The evaluation datasets consist of very common objects/animals which are well-represented in the training data of diffusion models such as cars and birds. It would help understanding the strenghts and weaknesses of the method to see its performance on a dataset with less common meta-classes, for example the iNaturalist.”**
> - We need to differentiate between two cases:
>   - **Common Meta Classes with Uncommon Subclasses:** While the meta classes like airplanes and birds are common, the subclasses are not. For instance, generating in a text-to-image manner a Boeing 737-300 airplane can be challenging (see Figure 1 in the rebuttal PDF).
>   - **Uncommon Meta Classes:** Some datasets we chose have uncommon meta classes. For example:
>     - DTD (Describable Textures Dataset): Contains specific textures such as Blotchy, Honeycombed, and Laced, which are not very common.
>     - CompCars: Includes images of car parts such as headlights and taillights, which are not commonly associated with the correct caption in datasets like LAION, that were used to train the generation models.
>
> We realize that evaluating on even more uncommon object datasets could further demonstrate the robustness of our method, and if the reviewer finds that it will help the exposition, we will have results for a uncommon dataset for the camera-ready version of our paper.
>
> [1] Rao, Yongming, et al. "Counterfactual attention learning for fine-grained visual categorization and re-identification." Proceedings of the IEEE/CVF international conference on computer vision. 2021.

---

> > ### Comment · Reviewer_JRLS · 2024-08-11
> >
> > Thank you for the response and your effort!
> >
> > My concerns have been mostly addressed and I am keeping my positive score. The only remaining one is that I am not convinced the DTD dataset results are enough to show the generalization of the method to metaclasses less frequently represented in the training data of diff. models.
> >
> >  As mentioned in the original review, I think this is a well-written paper with solid contributions. To make the contributions stronger, I am missing positive results on more diverse datasets, for example, showing the broader applicability of the method outside of FGVC, or more datasets with less common objects, to increase my score.

---

> > > ### Author Response · Authors · 2024-08-14
> > >
> > > We are happy that most of your concerns are resolved, and we thank you for your positive feedback.

---

### Official Review · Reviewer_yjvN · 2024-07-15

**Soundness:** 2
**Presentation:** 3
**Contribution:** 3
**Rating:** 6
**Confidence:** 4

**Summary:**

This paper presents SaSPA, a generative augmentation method specifically designed for FGVC. This method generates diverse, class-consistent synthetic images through conditioning on edge maps and subject representation. They use ControlNet condoned on edge maps and uses blip diffusion for its ability to zero-shot image generation. Their method is really smart and novel. They show results on Aircraft, Stanford Cars, CUB, DTD and Compcars for fine-grained evaluation.

**Strengths:**

1) I found the methodology of this paper to be highly impressive. The approach is both intelligent and relatively straightforward, making it intuitively appealing.
2) Utilizing GPT-4 to generate captions, followed by employing ControlNet on edges to create images via BLIP Diffusion, is a particularly smart strategy.
3) Additionally, the analysis on mitigating contextual bias is excellent, providing insightful and valuable results.

**Weaknesses:**

1) Datasets are of small scale on which the experiments have been done. I would expect this method to perform well on fine-grained evaluation of imagenet dogs setup as well. Showing experiments on 200 classes of dogs on ImageNet will make the paper even stronger.
2) How much is the performance improvement over methods like CLIP in few shot settings? Even if there is not much ( which is fair since here the aim is not image text training), it would be a good idea to show how behind this would be.
3) To be really honest, I believe the results are pretty incremental. In Table 3 the difference between Diff-Mix & SaSPA is 0.3%, which is quite small and is probably in the noise range. The method is really smart and novel and I think just using this method on small-scale experiments on fine-grained datasets is kind of underselling the paper. I believe the method has a lot of potential and can be used to do large-scale training. Especially using edge-based control net and blip diffusion is a really good idea.
4) I will urge the authors to rethink the experimental results and encourage them to show these results on larger datasets with more convincing results.
5) The Fig4 caption is really small and should be a bit more detailed.

**Questions:**

I think the paper needs more experimental results, but the image generation is really good.

**Limitations:**

Yes.

---

> ### Author Rebuttal · Authors · 2024-08-06
>
> Thank you for your feedback. We appreciate your recognition of the novelty of SaSPA. We're glad you found our strategy and analysis impressive and valuable.
>
>
> **“Datasets are of small scale on which the experiments have been done. I would expect this method to perform well on fine-grained evaluation of imagenet dogs setup as well. Showing experiments on 200 classes of dogs on ImageNet will make the paper even stronger.”**
> * Small-scale datasets are common in FGVC due to the infrequent appearance of specific objects in the real world and the subtle differences between sub-categories, which complicate labeling efforts [1]. This limited data availability is precisely what makes FGVC an interesting topic, as it highlights the challenge and importance of generating effective training data.
> * The Stanford Dogs dataset (also referred to as ImageNet Dogs) contains approximately 20,000 images across 120 classes. This is comparable in scale to the Stanford Cars dataset, which contains about 16,000 images across 196 classes. Thus, the scale of our current datasets is consistent with standard practices in FGVC research.
> * Nevertheless, to further address the concern and increase confidence in our method, we conducted an experiment on the Stanford Dogs dataset, as requested. The results are presented in Table 2 of the rebuttal PDF. Additionally, we included the Oxford-IIIT Pet Dataset in our evaluation. Both datasets showed improved results with SaSPA.
>
>
>
>
>
> **“How much is the performance improvement over methods like CLIP in few shot settings? Even if there is not much ( which is fair since here the aim is not image text training), it would be a good idea to show how behind this would be.”**
>
> The zero-shot accuracy of CLIP is included in Figure 2 of the rebuttal PDF. As shown, SaSPA consistently outperforms CLIP in all shots, including 4-shots, whereas other augmentation methods fall short for Cars and DTD.
>
>
>
> **“To be really honest, I believe the results are pretty incremental. In Table 3 the difference between Diff-Mix & SaSPA is 0.3%, which is quite small and is probably in the noise range. ”**
>
> * Our main results (Table 1, Table 2, Figure 4) show significant improvements over recent SoTA augmentation methods, both generative and non-generative, across various setups.
> * As for Table 3, diff-mix is a **concurrent paper**. ​​According to the official guidelines, authors are not expected to compare to such works. Nevertheless, we chose to add this comparison for completeness. The methods are quite different, and we believe both are valuable for the research community. Unlike Diff-Mix, which uses fine-tuning for its generative model, our method does not rely on such heavy fine-tuning. We have made it clearer in the paper that diff-mix is a concurrent work, as this point may have been missed.
>
>
>
> **“The method is really smart and novel and I think just using this method on small-scale experiments on fine-grained datasets is kind of underselling the paper. I believe the method has a lot of potential and can be used to do large-scale training. Especially using edge-based control net and blip diffusion is a really good idea”**
>
> Thanks for acknowledging the novelty of our method!
>
>
>
> **“I will urge the authors to rethink the experimental results and encourage them to show these results on larger datasets with more convincing results.”**
>
> See response to W1 and W3 above. We added new results.
>
>
> **“The Fig4 caption is really small and should be a bit more detailed.”**
>
> Thank you for the feedback. We have updated the caption for Figure 4 to be more detailed and easier to read.
>
>
> [1] Dunlap, Lisa, et al. "Diversify your vision datasets with automatic diffusion-based augmentation." Advances in neural information processing systems 36 (2023): 79024-79034.

---

### Author Rebuttal · Authors · 2024-08-06

We appreciate the reviewers' time, thoughtful comments, valuable suggestions, and their recognition of the potential positive impact of our method. Below, we address their common questions and concerns, in addition to the individual response per-review. In response to the feedback received, we have made several modifications to the paper and added new results, as requested. These changes have increased the clarity of our work and strengthened our findings even more.


## More Datasets

As requested by some reviewers, we added evaluations on two new datasets: the Stanford Dogs dataset and the Oxford-IIIT Pet Dataset.

* The Stanford Dogs dataset contains 20,580 images of 120 dog breeds from around the world, built using images and labels from ImageNet for fine-grained visual classification. We used 50% of this dataset due to time constraints.

* The Oxford-IIIT Pet Dataset contains 7,349 images of 37 breeds (25 dog breeds and 12 cat breeds) and is also used for fine-grained visual classification. We used 100% of this dataset.

* In Table 2 of the rebuttal PDF, we compare SaSPA with CAL-Aug, which is typically the strongest traditional augmentation in our experiments.
* We observed that SaSPA results in an improvement for both datasets, further strengthening the findings in our paper regarding SaSPA as a generative augmentation method.
* Combined with the results on DTD and CUB, we now have four datasets that are not of rigid objects, demonstrating that our method is effective beyond rigid objects (a concern raised by reviewer Z7K4)

## Will SaSPA Work on Higher Resolutions?
We have conducted additional experiments using 448x448 resolution on CompCars, DTD, and CUB datasets, employing SaSPA and the best augmentation method for each dataset (as per Table 1 in our paper). Each experiment was repeated with two seeds.

Table 1 in the rebuttal PDF presents the test accuracy for these high-resolution runs. Combined with our previous diff-mix comparison which used both 448x448 and 384x384 resolutions (Table 3 in the paper), we now have higher resolution results for all datasets.

Key findings:
* We observed consistent improvements across all datasets except CUB.
* For CUB, we hypothesize that the very fine-grained details such as feather patterns and colors present a significant challenge for generative methods, making it difficult to generate accurate representations at higher resolutions.

In conclusion, SaSPA demonstrates promising results for most datasets, affirming its overall benefits.

---

### Decision · Program_Chairs · 2024-09-25

**Decision:**

Accept (poster)

**Comment:**

The paper introduces a generative augmentation method for fine-grained visual categorization (FGVC) that creates diverse, class-consistent synthetic images using ControlNet conditioned on edge maps and BLIP diffusion for zero-shot image generation. The method shows promising results on datasets like Aircraft, Stanford Cars, CUB, DTD, and Compcars.

All the reviewers are satisfied with the authors' concerns. The authors need to fix the minors concerns, including the need for a limitations section addressing issues like less satisfying performance on high-resolution datasets (e.g., CUB) and demonstrating broader applicability by testing on more diverse datasets beyond FGVC.